A comparative study of stereopsis measurements: analyzing natural conditions versus dichoptic presentation using smartphones and ultraviolet printer technology

Liu Lu
Liu Jiang
Xu Lingxian
Zhao Lingzhi
http://orcid.org/0000-0001-6167-5659 Wu Huang wuhuang@jlu.edu.cn
Department of Optometry, The Second Hospital of Jilin University , Changchun , China
Redondo Beatriz
Electronic publication date: 2024 Feb 12
Publication date: 2024
Volume: 12
Electronic Location ID: e16941
Received 2023 Aug 17; Accepted 2024 Jan 23
Copyright: © 2024 Liu et al.
Copyright year: 2024
Copyright holder: Liu et al.
License: This is an open access article distributed under the terms of the Creative Commons Attribution License, which permits unrestricted use, distribution, reproduction and adaptation in any medium and for any purpose provided that it is properly attributed. For attribution, the original author(s), title, publication source (PeerJ) and either DOI or URL of the article must be cited.
License URL: https://creativecommons.org/licenses/by/4.0/

Keywords: Natural state, Dichoptic, Visual acuity, Stereopsis, Smartphones

Funding: Jilin Provincial Science & Technology Department, China 20230203100SF This work was supported by the Jilin Provincial Science & Technology Department, China (No. 20230203100SF). The funders had no role in study design, data collection and analysis, decision to publish, or preparation of the manuscript.

==============================
Background

Accurate differentiation between stereopsis assessments in the natural and dichoptic presentation states has proven challenging with commercial stereopsis measurement tools. This study proposes a novel method to delineate these differences more precisely.

Methods

We instituted two stereopsis test systems predicated on a pair of 4K smartphones and a modified Frisby Near Stereotest (FNS) version. Stereoacuity was evaluated both in the natural environment state (via the modified FNS) and the dichoptic state (via smartphones). Thirty subjects aged 20 to 28 years participated in the study with the best-corrected visual acuity (VA) of each eye no less than 0 logMAR and stereoauity of no worse than 40″. Varying degrees of monocular VA loss were induced using the fogging method, while this study does not explore conditions where the VA of both eyes is worse than 0 logMAR.

Results

When the VA difference between the two eyes did not exceed 0.2 logMAR, the modified FNS produced lower stereoacuity values compared to the 4K smartphones (Wilcoxon signed-rank test: difference = 0 logMAR, Z = −3.879, P < 0.001; difference = 0.1 logMAR, Z = −3.478, P = 0.001; difference = 0.2 logMAR, Z = −3.977, P < 0.001). Conversely, no significant differences were observed when the binocular vision difference exceeded 0.2 logMAR (difference = 0.3 logMAR, Z = −1.880, P = 0.060; difference = 0.4 logMAR, Z = −1.784, P = 0.074; difference = 0.5 logMAR, Z = −1.812, P = 0.070).

Conclusion

The findings suggest that stereoacuity values measurements taken in the natural environment state surpass those derived from the dichoptic presentation. However, the observed difference diminishes as stereopsis decreases, corresponding to an increase in induced anisometropia.

Introduction

Stereopsis is a crucial binocular function that assists people in accurately judging distance, whereas stereoacuity evaluates the discernible changes in depth and can be assessed using various methods (Nabie et al., 2019). Such evaluations may be dichotomized into two distinct categories, depending on whether the presentation of stimuli is dichoptic during the examination process (Howard & Rogers, 2012).

The first category encompasses measurements taken within the natural state. The natural state implies that from the perspective of the test subject, what is perceived is genuine and unassisted by additional instruments, with the exception of routine refractive corrective eyewear. However, due to the interpupillary distance (IPD), the right eye perceives slightly more of the right side of the visual target, and the left eye perceives slightly more of the left side. This variation gives rise to disparity. When individuals with differing IPD view the same test target from an identical test distance, the disparity varies. The natural state measurements necessitate a minimum of three parameters: disparity, test distance, and IPD. This encompasses techniques such as the Howard Dolman apparatus and the Frisby Near Stereotest (FNS) (Bohr & Read, 2013). Conversely, the second category pertains to measurements conducted in a dichoptic presentation state. In this situation, the visual stimuli observed by the right eye are inaccessible to the left eye and vice versa. This condition can be implemented through the utilization of polarized glasses (Read & Bohr, 2014), as exemplified in the Titmus Fly Test; red and green glasses (Bonfanti et al., 2021), such as in the TNO Stereotest; or via the employment of naked eye 3D technology, as demonstrated in the Lang Stereotest.

Some researchers have focused on the comparative outcomes of examinations conducted within the natural state environment as opposed to those performed in the dichoptic state. Simon’s previous study demonstrated lower examination values from the FNS compared to the Random-Dot E Stereotest (RDE) and TNO (Simons, 1981). Simons attributed this difference to the presence of more pronounced monocular cues in the FNS test, with one reported case demonstrating an adult passing three test plates with only one eye successfully. Anketell, Saunders & Little (2013) evaluated stereoacuity with FNS and TNO in 186 school-aged children, observing lower FNS results in the middle school group compared to the primary school group (Mann-Whitney test, crossed disparity: Z = 4.67, P < 0.0001; uncrossed disparity: Z = 4.67, P < 0.0001), but identified no significant differences between the TNO groups (Z = 1.35, P = 0.18). The FNS stereoacuity values were significantly lower than those of TNO, regardless of whether the disparity was crossed or uncrossed (crossed disparity: Z = −11.34, P < 0.0001; uncrossed disparity: Z = −1.067, P < 0.0001). Ateiza & Davis’s (2019) examination of anisometropic amblyopia utilizing FNS and TNO revealed that stereopsis in the FNS group was markedly superior compared to the TNO group for both amblyopia and control groups. The authors hypothesized that FNS and TNO might evaluate distinct facets of stereo vision. Further, an analysis by Biddle, Hamid & Ali (2014) ascertained that subjects demonstrated superior stereoacuity on the FNS test when contrasted with TNO and Titmus. Consequently, some researchers suggest that stereoacuity values measurements obtained in the natural state are lower than those procured under dichoptic presentation (Chung, Park & Shin, 2017; Fu, Birch & Holmes, 2006; Hall, 1982; Hartmann et al., 2015; Heron et al., 1985).

Nonetheless, the level of comparability between divergent inspection methodologies is found wanting. This is attributable to various factors, such as the discrepancy in the design of test graphics, the range of disparity steps, the extremity values of disparity, and potential monocular cues, all of which potentially confound comparative results.

In the natural state, FNS facilitates the continuous quantification of the stereoacuity threshold. This assessment is predicated on variables such as the thickness of the test plate, the distance of the test, and the subject’s IPD. The results of the FNS could potentially span the entire measurement range. In practical applications, examiners typically resort to thresholds of 340″, 170″, and 85″, ascertained based on a 65 mm IPD at a fixed distance of 40 cm (Schneck et al., 2000; Wright & Wormald, 1992).

In contrast, the quantitative examination thresholds enlisted in the most recent edition (19th edition) of the TNO Stereotest comprise of 480″, 240″, 120″, and 60″, with the previous edition incorporating additional thresholds of 30″ and 15″. For the Titmus Stereotest, the quantitative examination thresholds encompass 800″, 400″, 200″, 140″, 100″, 80″, 60″, 50″, and 40″. As for the Random Dot Stereoacuity Test (RD), the quantitative thresholds include 400″, 200″, 160″, 100″, 63″, 50″, 40″, 32″, 25″, 20″, 16″, and 12.5″.

In situations where the subject’s true stereoacuity stands at 10″, the results measured by FNS, TNO, Titmus, and RD would be 85″, 60″, 40″, and 12.5″, respectively, implying a hierarchical order of FNS > TNO > Titmus > RD. This scenario can be attributed to the differing extreme values associated with each measurement methodology. Conversely, for an individual with a stereoacuity of 160″, the results derived from FNS, TNO, Titmus, and RD would be 170″, 240″, 200″, and 160″, respectively, thus suggesting a hierarchy of TNO > Titmus > FNS > RD. This situation stems from the different step ranges adopted by each of these measurement techniques. Additionally, the results of the examination could also be influenced by variations in the form of visual targets, such as those based on contours or random dots (Chopin et al., 2021; Wong, Woods & Peli, 2002).

It is evident that there exist pronounced discrepancies across various commercial tools for stereopsis measurement. Consequently, these variations yield inconsistent stereoacuity values when derived from differing evaluation methodologies, engendering difficulties in rendering accurate comparisons. This study aims to precisely compare stereopsis tests performed in both dichoptic and natural states. VA significantly influences stereopsis (Atchison et al., 2020). A decrease in VA, whether in one eye or both (Al-Qahtani & Al-Debasi, 2018), can markedly affect stereoacuity. To explore this further, we reduced monocular VA using the fogging method to induce a difference in VA between the two eyes, reducing stereopsis. This study not only compared stereoacuity in natural and dichoptic environments with normal VA but will also examine how stereopsis changes in these environments after inducing varying degrees of monocular VA loss. The approach aims to provide a more comprehensive understanding of the differences in stereopsis between natural and dichoptic environments. As such, we have chosen to eschew commercial stereopsis measurement tools in favor of a novel test system utilizing smartphones and a phoropter for the dichoptic presentation state, which maintains consistency in critical variables such as the size and form of the patterns, the range of test steps, extreme values, and test distances. For tests in the natural state, modified FNS plates were utilized to align with the dichoptic test system. The objective is to probe with greater academic rigor whether differences exist in the value of stereopsis when examined in the natural state versus a dichoptic state.

Materials and Methods

Test system

The dichoptic presentation test system

The stereopsis test system under investigation is premised on the utilization of a phoropter (TOPCON VT-10; TOPCON Corp, Tokyo, Japan) in conjunction with two 4K smartphones (Sony Xperia XZ Premium; Sony Mobile Communications Inc., Tokyo, Japan). The test distance was calibrated to 65 cm, such that a one-pixel distance epitomizes a 10″ disparity. In an effort to facilitate fusion, two approximately 5.5ΔBO Risley prisms were positioned in front of both eyes (Fig. 1A) (Liu et al., 2021; Xu, Liu & Wu, 2022).

Figure 1 Photograph of the testing system.

(A) The photograph of two 4K smartphones systems. (B) The photograph of a modified Frisby test system. The acrylic plate was fixed using four magnetic stripes on a modified near-test plate. Another near-test plate, covered with a piece of white paper, was fixed behind the acrylic plate to avoid possible monocular cues.

The configuration of the test graph was modelled after that of the FNS (Stereotest Ltd., Sheffield, UK), albeit with modifications to accommodate the display capabilities of the smartphone. Thus, the dimensions of the images were scaled down to a quarter of the original pattern. This modification resulted in each test square frame being transformed from a 6 × 6 cm format to a more compact 3 × 3 cm arrangement. The range of disparities spanned from 200″ to 10″, with a total of 20 pairs of test charts, each separated by an interval of 10″. Cross-disparity was a consistent feature throughout the entirety of the test (Fig. 2).

Figure 2 Legend of the test pattern of the smartphones test system.

(A) As seen by the right eye; (B) as seen by the left eye; (C) as the simulation of the perception of the test images. The upper left prominent square was designated as the target symbol. The disparity of this page was set at 120″.

The natural state test system

For testing within the natural environment state, an 8 × 8 cm acrylic plate was utilized to simulate FNS. All patterns were retained; however, their dimensions were reduced to a quarter of the original size. Consequently, each square frame measured 3 × 3 cm, mirroring the dimensions displayed on the smartphone. A UV (ultraviolet) printer (Sonpoo 3060 printer; Shenzhen Songpu Industrial Group Co., Ltd., Shenzhen, China) was employed for dual-sided printing. The stereoscopic mark (circle) was printed on one side of the acrylic plate, with the control pattern printed on the reverse. The meticulous alignment of the patterns on both sides was a prerequisite during the printing process. A total of 20 acrylic plates were utilized, with thicknesses spanning from 0.5 to 10 mm, and a variation step range of 0.5 mm.

The modified FNS plate was securely positioned using four magnetic strips on an adapted near-test plate (Fig. 1B). The distance can be meticulously fine-tuned on the near-vision rod to conform to the subject’s IPD, thus satisfying the requirements for measurement within the natural state.

It’s important to clarify that in this study, the ‘natural state’ is contrasted with the state of dichoptic vision. Dichoptic state refers to a scenario where the images perceived by the right and left eyes are distinct; the object visible to the right eye is not visible to the left eye, and vice versa. In this case, the disparity is set horizontally, concealed within the images presented to each eye. On the other hand, the ‘natural state’ implies that the same object is simultaneously visible to both the right and left eyes, allowing for the perception of depth as it exists in reality.

Participants

The study comprised a cohort of thirty subjects (11 males and 19 females) aged 20 to 28 years. A prerequisite for inclusion stipulated that the best-corrected VA for each eye could not be less than 0 logMAR. Moreover, participants’ stereoacuity had to be a minimum of 40″ as measured by the Fly Stereopsis Test (Stereo Optical Company, Inc., Chicago, IL, USA). Before participation in the study, a written, informed consent was obtained from all participants. The research protocol underwent a thorough ethical review and was subsequently approved by the Ethics Committee of the Second Hospital of Jilin University (Approval No. 2022-260).

Test method

Determination of visual condition

The refraction of each subject was ascertained utilizing a standardized optometry process. This entailed positioning the near-vision rod and fixing a 4K smartphone at a distance of 65 cm. An E-symbol VA chart was presented on the smartphone display before the subject, with the optotype size, optotype count, and progression matched that of the ETDRS VA chart (Fig. 3). The VA chart is designed in accordance with the Bailey-Lovie design principles (Pease, 2006). Its distinctive feature is a tumbling ‘E’ in four orientations—up, down, left, and right. This specific design choice ensures equal average legibility across all optotypes. Each row of the chart comprises five optotypes, with spacing between the letters and the rows proportionate to the letters’ size. Furthermore, the chart exhibits a logarithmic size progression across all lines. During the VA examination, the chart is systematically checked line by line, starting from the top and moving downwards. A line is successfully completed if the participant correctly identifies three or more visual targets. This criterion is applied consistently to determine the successful recognition of each line on the chart. The visual angle for each line of symbols was calculated based on the fixed distance of 65 cm. In the case of all participants with normal distance VA and accommodation, none of the VA measurements taken at 65 cm fell below 0 logMAR.

Figure 3 Visual acuity chart for a test at 0.65 m expressed on a 4K smartphone.

The left eye was covered through the phoropter, and positive lenses were placed in front of the right eye. The highest degree of positive lenses (or the lowest degree of negative lenses) was recorded when the VA reached 0.6 logMAR. Subsequently, negative lenses (or reduced positive lenses) were placed in front of the right eye, and the diopter was recorded when the VA reached 0.5 logMAR, and so forth. This unfogging procedure continued, documenting the diopter at each stage when the VA reached 0.4, 0.3, 0.2, 0.1, and 0 logMAR. If a 0.25DS change potentially caused a shift in two lines of VA during the unfogging procedure, 0.12DS auxiliary lenses of the phoropter could be used. This procedure yielded six diopters corresponding to 0.5 to 0 logMAR.

Following this, uncover the left eye, and the right eye was covered, lenses were added in front of the left eye, and the maximum degree of positive lenses (or the lowest degree of negative lenses) was recorded when it reached 0 logMAR. A flowchart delineating the procedure for determining the visual condition was illustrated in Fig. 4.

Figure 4 The flow chart of the test procedure.

Determination of test distance

The test distance for smartphone usage was uniformly set at 0.65 m. In contrast, the test distance measured within the natural environment was derived based on the subject’s IPD. In this study, specific IPD values were correlated with their corresponding test distances. Specifically, IPD of 55 and 56 mm corresponded to a test distance of 0.62 m, 57 mm and 58 mm to 0.63 m, 59 mm and 60 mm to 0.64 m, 61 mm to 0.65 m, 62 mm and 63 mm to 0.66 m, 64 mm and 65 mm to 0.67 m, and 66 mm and 67 mm to 0.68 m. At each of these specified distances, a test plate thickness of 0.5 mm represented a disparity of 10″. For every incremental increase of 0.5 mm in plate thickness, the disparity increased by 10″, up to a maximum of 10 mm (200″).

Alterations to the test distance exerted an impact on the visual angle. Nevertheless, the visual angle ascertained within the range of 0.62 to 0.68 m remained consistent with the angle determined at 65 cm (the difference in equivalent visual angles at distances of 0.62 and 0.68 m amounts to a mere 0.04 logMAR). Thus, a difference in distance of 60 mm was deemed inconsequential to the judgement of VA within the context of this experiment.

Procedure for stereopsis test

Evaluation of stereoacuity via smartphones

For the examination, the lenses corresponding to the VA of 0 logMAR at 0.65 m were positioned in front of the left eye, while those achieving 0.5 logMAR at the same distance were positioned before the right eye. The test commenced at 200″. Participants were directed to select a stereoscopic symbol from a choice of four-square frames. Should it prove truly impossible to ascertain despite diligent efforts, conceding defeat is permissible. In cases where participants failed the 200″ test, a result of 240″ was recorded. Conversely, if the participant made the correct choice, the disparity was decreased by 10″, a 190″ test was then conducted, and the procedure was repeated until the subject made an incorrect choice. The last correct response was recorded as the participant’s stereopsis threshold.

The lenses in front of the left eye remained consistent, while those corresponding to 0.4 logMAR at 0.65 m were placed before the right eye for subsequent evaluation. Six groups were categorized based on the VA difference between the left and right eye (Group 1: no difference; Group 2: difference = 0.1 logMAR; Group 3: difference = 0.2 logMAR; Group 4: difference = 0.3 logMAR; Group 5: difference = 0.4 logMAR; Group 6: difference = 0.5 logMAR). From this process, six different stereoacuity results of each participant were acquired.

Evaluation of stereoacuity using the modified FNS test

The test distance was established based on the individual subject’s IPD. The same test procedure was used for the smartphone, spanning from 200″ (plate thickness of 10 mm) to 10″ (plate thickness of 0.5 mm), with a step range of 10″ (plate thickness of 0.5 mm). The categorization of test followed the same schema as for the smartphone tests. However, in contrast to the smartphone assessments, an additional monocular evaluation was incorporated. At the end of each stereoacuity determination, two supplemental tests were conducted at the threshold of stereopsis (right eye only and left eye only) to prevent false positive results due to monocular cues. A flowchart outlining this process is depicted in Fig. 4.

The sequence of tests utilizing smartphones and the modified FNS was randomized.

Statistical analyses

PASW Statistics 26.0 (SPSS Inc., Chicago, IL, USA) processed the data. The nonparametric Wilcoxon signed-rank test was applied for data that did not follow the normal distribution. A significance level of P < 0.05 was used for statistical significance.

Results

The mean age of the thirty subjects (11 males and 19 females) was 27.00 ± 4.24 years. The best-corrected VA for each eye is no less than 0 logMAR, and the median (interquartile range) of the Fly Stereopsis Test was 28.5″ (8.25″), of the modified FNS was 20″ (10″), of the smartphone was 30″ (10″). The outcomes of the Wilcoxon signed-rank test are presented in Table 1. For Groups 1 to 3, where the interocular VA differential was 0, 0.1, and 0.2 logMAR respectively, a statistically significant difference was identified between assessments conducted via smartphones and the modified FNS. Conversely, in Groups 4 to 6, where the interocular VA differential was 0.3, 0.4, and 0.5 logMAR respectively, no significant difference was detected between the smartphone and modified FNS results. A boxplot depicting this data is illustrated in Fig. 5.

Table 1 Median (interquartile range) of the test data.

	Modified FNS
M (QR)	Smartphone
M (QR)	Wilcoxon signed-rank test	
Z	P	
Group 1	20 (10)	30 (10)	−3.879	<0.001	
Group 2	25 (15)	40 (40)	−3.478	0.001	
Group 3	40 (40)	50 (65)	−3.977	<0.001	
Group 4	75 (85)	85 (110)	−1.880	0.060	
Group 5	115 (105)	150 (165)	−1.784	0.074	
Group 6	185 (127.5)	200 (105)	−1.812	0.070	
Note:

Group 1, the difference of visual acuity between binocular was 0 logMAR; Group 2, the difference of visual acuity between binocular was 0.1 logMAR; Group 3, the difference of visual acuity between binocular was 0.2 logMAR; Group 4, the difference of visual acuity between binocular was 0.3 logMAR; Group 5, the difference of visual acuity between binocular was 0.4 logMAR; Group 6, the difference of visual acuity between binocular was 0.5 logMAR.

Figure 5 Boxplots of stereoacuity tested with the modified Frisby and smartphone tools.

Group 1, the difference of visual acuity between binocular was 0 logMAR; Group 2, the difference of visual acuity between binocular was 0.1 logMAR; Group 3, the difference of visual acuity between binocular was 0.2 logMAR; Group 4, the difference of visual acuity between binocular was 0.3 logMAR; Group 5, the difference of visual acuity between binocular was 0.4 logMAR; Group 6, the difference of visual acuity between binocular was 0.5 logMAR. The line perpendicular to the whisker above the box represents the maximum value, the upper edge of the box represents the third quartile, the thick line in the box is the median, the lower edge of the box represents the first quartile, the line perpendicular to the whisker below the box represents the minimum value, and the stars and circles represent extreme values.

In all six test groups, no participant achieved a passing score on the monocular assessment.

Discussion

The FNS and the Frisby Davis Distance stereotest (FD2) are widely employed methods for conducting measurements in the natural state, for near and far distances, respectively (Bohr & Read, 2013; Read et al., 2016). Conversely, numerous stereotests are conducted using the method of dichoptic presentation. Three dichoptic techniques are commonly employed in clinical settings: the red and green glasses method (e.g., TNO stereotest), polarized glasses method (e.g., Titmus stereotest), and the naked-eye 3D technology method (e.g., Lang stereotest). Although the computation of stereopsis is disparity-based, the form in which disparity is set varies between these two methodologies. In the dichoptic presentation state, the disparity is set horizontally, whereas it is established in the fore-and-aft direction in the natural environment test. The relatively low consistency among various commercially available stereopsis measurement methods complicates determining the equivalency of the results derived from these two types of methods.

Applying electronic stereoscopic measurement methods under a dichoptic state can enhance the comparability of different inspection methods. The test comparability can be improved by ensuring the same test range, test steps and eliminating other confounding factors. Wu, Liu & Wang (2018) compared the consistency of stereoacuities inspection using dual 4K smartphones with the traditional Fly Stereo Acuity test. They found a high level of agreement between the two methods. Moreover, electronic stereoscopic measurement can be integrated with various advanced devices, such as eye trackers, facial trackers, etc., to realize accurate, convenient stereopsis tests. Liu et al. (2023) combined stereopsis measurement with the eye tracker to create a new test method that achieves the self-service stereotest that can output objective indicators and is highly consistent with the conventional test methods. In this study, our results are in accordance with the experimental results mentioned by Langer & Siciliano (2015), who set out an experiment that employed a 3D display with blur generated by image processing to investigate the influence of blur cues on depth discrimination from disparity. The experiment was carried out using a Dell Precision M6700 laptop computer. PsychoPy (Peirce, 2007) was used to create and regulate the stimuli. To produce a dichoptic presentation, stereo images were viewed using NVidia’s 3D Vision shutter glasses. The viewing distance was 63 cm. They found that blur may be viewed as a weak cue to depth discrimination, consistent with our results that a mild decrease in monocular VA shows no significant effect on stereopsis.

In applying the FNS test, the potential influence of monocular cues is a primary concern. The test involves a target and contrast pattern on an acrylic plate, creating disparity through its thickness. Factors like lighting, background, plate positioning, and the examinee’s head position can inadvertently provide clues due to shadows or slight target movements. Maintaining a steady head position is crucial to avoid motion parallax, which could give extra cues. Therefore, the test requires a uniform background and precise alignment of the subject’s line of sight with the test plate. Despite efforts to reduce monocular cues, completely eliminating them is challenging, leading some researchers to question the FNS’s effectiveness in stereopsis examination (Letourneau, Beaulne & Duplessis, 1992; Moganeswari et al., 2015; Ohlsson et al., 2001; Simons, 1981).

In our study, we instituted several controls to minimize the influence of monocular cues. The placement of the plate was affixed to the near-vision rod of the phoropter, standardizing the lighting and the angle of the test plate to eliminate potential discrepancies due to shadowing and tilt. The observation hole of the phoropter confined head movement, while participants were consistently reminded to remain stationary throughout the examination. Following the test, an additional test was made under monocular conditions, and none of the participants passed this monocular test. Notably, while FNS features three test plates, our adaptation in this experiment includes 20. FNS’s variable inspection distance, tailored to the subject’s IPD, differs from our fixed inspection distance approach, marking a departure from the traditional FNS methodology. Additionally, given the extensive duration of our experiment’s measurement process and to avoid the consequent subject fatigue, we only presented the test plate once at each level, which, aligned with many stereotests, was a 4-choose-1 method adopted. Given the context of our experimental design, we did not perform a test-retest reliability analysis. Instead, our test plates have been set many enough; the test steps have been set small enough, and we aim to mitigate these inherent inaccuracies by minimizing the test steps. Furthermore, we selected young individuals with good responsiveness as our subjects to reduce the variability in the test results.

Despite our best efforts to eliminate monocular cues, significant differences were observed between the stereopsis test results obtained in the natural state and those gathered under the dichoptic state. This suggests that binocular cues might play a pivotal role in such settings (Chopin et al., 2019; Linton, 2020). Specifically, binocular cues may provide auxiliary support when the VA is approximately equivalent between the two eyes. However, this auxiliary effect dissipates when the discrepancy in VA between the two eyes exceeds two lines (0.2 logMAR).

Vertical disparity represents one form of binocular cue employed to discern depth perception. However, for vertical disparity to have a substantial influence on depth perception, the object under consideration must be positioned on the fronto-parallel plane and occupy a field of vision of at least 20 degrees (Rogers & Bradshaw, 1995). In our experiment, the scaled-down Frisby plate, reduced 3/4 its original size, was positioned approximately 0.6 m away from the participant and occupied a small portion of the visual field. Therefore, the impact of vertical disparity on judgement outcomes can reasonably be disregarded.

Vergence serves as another significant binocular cue in depth perception. Theoretically, the absolute distance can be determined by the angle at which the eyes are set. By gauging the rotation angle of the eyes, the visual system can calculate the distance from the point of vergence to the eyes (Banks et al., 2016; Parker, Smith & Krug, 2016). Viguier, Clement & Trotter (2001) posited that vergence provides a reliable means for ascertaining the distance to the target, particularly within a visually accessible space. Given that vergence decreases with the tangent value of the distance, it is generally perceived that the effective range of vergence does not exceed 2 m (Mon-Williams & Tresilian, 1999). However, some researchers argue that vergence could exert influence over a more considerable distance (Brenner & van Damme, 1998).

In this study, the target was enclosed by a background pattern. Consequently, the retinal disparity mechanism played a principal role in depth judgement, while remaining difference judgements may be attributable to vergence. We postulate that during the examination, as the eyes perform saccades across the entire Frisby plate, the angle of vergence experiences abrupt changes at the boundary between the target and background patterns. These sudden shifts in vergence could serve as cues, furnishing depth information. When the difference in VA between the two eyes is relatively minor (between 0 and 0.2 logMAR), it allows vergence to contribute to binocular cues effectively. However, as the difference in VA widens (exceeding 0.3 logMAR), the depth cues provided by vergence become less effective. The results mentioned are based on conditions of induced vision loss and may not align with outcomes observed under actual circumstances.

However, our study is not without its limitations. First, the size of the FNS was reduced to a quarter of its original size, which has not been tested for its effect on actual FNS results, thus limiting the comparability with other studies. Second, the alteration in VA was induced via fogging, which may not accurately reflect the authentic VA in daily life. Third, the better VA was constrained to 0 logMAR, thus the comparative results primarily reflect conditions wherein one eye’s VA is at 0 logMAR with the other eye having equal or poorer VA. This study does not explore conditions where the VA of both eyes is worse than 0 logMAR, and the levels of monocular blur are very close together; given that VA test-retest variability can exceed 0.1 logMAR, the large number of steps with small VA changes may be the reason behind the lack of difference between some groups. Lastly, the participants selected for our study all possess good VA and stereoacuity, which introduces a selection bias in our population sample.

Conclusions

The measurement of stereoacuity in the natural environment state yielded significantly lower values compared to those derived from the dichoptic presentation state. However, the observed difference diminishes as stereopsis decreases, corresponding to an increase in induced anisometropia.

Supplemental Information

Supplemental Information 1 Test results of Smartphone-based stereopsis values (arcsec) under different vision acuity condition (log MAR).

Click here for additional data file.

Supplemental Information 2 Test results of modified FNS-based stereopsis values (arcsec) under different vision acuity condition (log MAR).

Click here for additional data file.

We are grateful to Professor Frisby for his outstanding design of the FNS, which has been instrumental in guiding our research and enabling its completion. We thank the Editor and the reviewers for their insightful suggestions and careful manuscript reading. We also thank the participants for their time and generosity in contributing to this study.

Additional Information and Declarations

Competing Interests

Author Contributions

Human Ethics

Data Availability

The authors declare that they have no competing interests.

Lu Liu conceived and designed the experiments, performed the experiments, authored or reviewed drafts of the article, and approved the final draft.

Jiang Liu performed the experiments, analyzed the data, authored or reviewed drafts of the article, and approved the final draft.

Lingxian Xu analyzed the data, prepared figures and/or tables, authored or reviewed drafts of the article, and approved the final draft.

Lingzhi Zhao analyzed the data, prepared figures and/or tables, authored or reviewed drafts of the article, and approved the final draft.

Huang Wu conceived and designed the experiments, authored or reviewed drafts of the article, funding acquisition, and approved the final draft.

The following information was supplied relating to ethical approvals (i.e., approving body and any reference numbers):

This study received approval from the Ethics Committee of the Second Hospital of Jilin University.

The following information was supplied regarding data availability:

The raw data are available in the Supplemental Files.

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
