# Peer review of "A comparative study of stereopsis measurements: analyzing natural conditions versus dichoptic presentation using smartphones and ultraviolet printer technology"

_PeerJ, doi:10.7717/peerj.16941_

## Round 0.1 · original submission · Major Revisions

It has been reviewed by three experts in the field. Revisions are
necessary before the manuscript is suitable for publication.

·

Basic reporting

Paragraph starting on line 53
When discussing whether one test is superior to another, you need to state what measure you are using to determine superiority eg is this in terms a higher stereoacuity score, a lower test retest variability or some other measure of superiority? If the test is simply giving a lower (meaning better) stereoacuity score, that does not necessarily mean the test is superior eg the score could be inaccurate if the test contains monocular cues. This paragraph needs to further explore the reasons for the differences as simply comparing stereoacuity results and concluding that the test with the better score means the test is better is flawed.
The authors have accurately described the impact of the minimum measurable stereoacuity plus acknowledged the impact of variations in step sizes but haven’t considered the impact of potential monocular cues, which is very important when evaluating the accuracy of these tests.
Line 94 – The sentence that starts in the middle of this line is unnecessarily complicated. I don’t understand the point the authors are trying to make. The aim should be clearly stated. In the following paragraph it is not clear that the novel test is made in both a dichoptic state and natural state.

It would be appropriate to include the paper - Bohr I, Read JCA. Stereoacuity with Frisby and Revised
FD2 stereo tests. PLoS One. 2013;8. as it is highly relevant to this paper. Also a forward citation search based on this paper by Bohr also other relevant publications that are currently not included.

Experimental design

Line 109 – need to provide details on the rationale for the use of 5.5 dioptre BO prisms. I understand you are simulating convergence but have not provided any detail on the choice of magnitude of prism

Line 130 – For the “natural state” condition the phoropter is used and the distance is fine-tuned based on the subject’s pupil distance (please change to inter pupillary distance), but given that the FNS is not usually presented in this way it is not truly in the natural state.
Line 138 – seems odd to choose the Fly stereo test for screening purposes when all the testing is based on the FNS.
Line 148 – a tumbling E test was used but then the authors state the orientation was set to the mode of the ETDRS chart, what does this mean? The orientation of the letter E can only be one of four options, which is not related to the ETDRS. Maybe they mean the optotype size and progression matched that of the ETDRS chart (following logMAR principles), but this needs clarifying.
Line 153 – this paragraph describes the fogging procedure but this has not been mentioned previously. The overall study aims should include something about the evaluation of the tests under different degrees of anisometropia and the rationale for doing so.
For this same paragraph it should also provide further details on the VA testing procedure, how many optotypes were presented at each level, how many optotypes did the participant need to correctly or incorrectly identify to determine the VA?
Line 178 – please provide the angles rather than simply stating there’s no impact of a 60mm change
Line 190 - states the last correct response was recorded as their threshold, does this mean each disparity level was only presented once? For a four alternative forced choice method, with a one in four chance of guessing correctly, only presenting each level once will increase the variability in the responses due to the increased chances of a lucky guess.
When the disparity was reduced on the smartphone test, was the position of the circle with disparity changed as well? I would assume so but this has not been stated, or what randomisation process was used to create the change.
For the modified FNS test was there any procedure used to determine the location of the disparity or was it based on examiner choice? Or, as the text suggests, was it purely based on distance and inter pupillary distance? If so there could be a large latent deviation that is being controlled meaning vergence is being exerted (and possible increased by the prisms, depending on the latent deviation) and not standardised between participants.
Line 209 – need to specify how the “necessary amount” of prism was determined. Did they neutralise the participants latent deviation? The authors refer to parallel visual axes, but this would not be appropriate for a near target where a degree of convergence is necessary to gain binocular single vision.
Statistical analysis – give the inclusion of an arbitrary figure of 240” when the participant could not identify the 3D shape, non-parametric analysis should be used throughout.

Validity of the findings

Results
Should start the results with the baseline demographics of the participants, the age is stated in the methods but then gives the inclusion criteria rather than their actual VA or stereoacuity, which should be reported in the results.
Figure 4 and table 1 show the same data, it is not necessary to have both, the figure is more informative.

Discussion
Again need to clarify the use of the term superior when discussing published data.
Line 288 – at the end I think the authors mean monocular cues rather than binocular
Lines 259-297 seem to be more suited to the introduction (and some aspects overlap with the introduction) as it’s just discussing the variations in the findings from different stereotests.
Line 305 – the authors evaluate the significant differences between the natural and dichoptic states, however they have simply fixed on the low p value without considering the magnitude of the differences. Given that the variance has been reported (eg see Bohr I, Read JCA. Stereoacuity with Frisby and Revised FD2 stereo tests. PLoS One. 2013;8.) to be greater than 10” then a median difference of 10” is not clinically significant.
Line 329 – the authors postulate that there are abrupt changes in vergence when changing fixation from the target to the background pattern, but the maximum plate thickness on the FNS is 6mm which would not necessitate a large change in vergence. Given that all participants had excellent stereoacuity, they would be using the thinner plates with even smaller changes so this theory does not seem to be applicable.
Line 340 – without factoring in the participants latent deviation and knowing how much vergence is being exerted then this conclusion cannot be made.
Other study limitations that also need addressing in the discussion are:
The levels of monocular blur are very close together, given that VA test-retest variability can exceed 0.1 logMAR the large number of steps with small VA changes may be the reason behind the lack of difference between some groups.
The ceiling effect has not been mentioned, looking at the data it suggests there were a number of participants who achieved the lowest score possible of 10”, meaning the authors did not obtain a true measurement of stereoacuity.

Conclusions
The conclusions are not valid without considering the points raised above.

Additional comments

As the test is modelled on the FNS are there any intellectual property issues related to the use of the principles of the FNS or was permission sought from Professor Frisby? There’s nothing mentioned in the acknowledgements.

·

Basic reporting

The manuscript titled “A comparative study of stereopsis Measurements: Analyzing Natural Conditions Versus Dichoptic Presentation Using Smartphones and Ultraviolet Printer Technology” was well written, clear and easy to understand. The structure and format are in conformity with PeerJ standards. Authors need to add DIO to the references if it is available. Figures are relevant, but they need basic labels added to them. The figure 1 requires a description, and a high-power view of testing images on the monitor can be added to it.

Experimental design

The study goal is clear, relevant, and a knowledge gap has been identified. A novel method is proposed to more precisely delineate the differences between the stereopsis assessments in the natural state and dichoptic presentation state. The method has been well explained in sufficient detail. Providing more detailed information will enhance the quality of this article.

Validity of the findings

Impact and novelty have not been assessed. Meaningful replication encouraged where rational & benefit to literature is clearly stated. All the underlying data has been provided; they are robust, statistically sound, and controlled. The conclusions are well aligned with the original research goal, but they require more details and a better statement.

Additional comments

The article is well written and clear. Authors need to add DIO to the references if it is available. Figures are relevant, but they need basic labels added to them. The figure 1 requires a description, and a high-power view of testing images on the monitor can be added to it. The method has been well explained in sufficient detail. Providing more detailed information will enhance the quality of this article. The conclusions are well aligned with the original research goal, but they require more details and a better statement.

Reviewer 3 ·

Basic reporting

This manuscript instituted two stereopsis test systems predicated on a pair of 4K smartphones and a modified version of the Frisby Near Stereotest (FNS) to differenciate stereopsis assessments in the natural and dichoptic presentation states. The method appears to be thorough and well-designed, taking into account various factors such as visual acuity, test distance, and categorization of participants. However, the sample size is relatively small making the results somehow unpersuative.

Experimental design

1. Only twenty participants was evaluated which is relatively too small .The sample size could be larger.
2. Why “Fly Stereopsis Test” is set in the inclusion criteria as this is not mentioned before?

Validity of the findings

The discussion section could bemore concise. Some of the content can be placed in the background section. And it is suggested that the authors include other electronic stereoscopic measurement methods under dichoptic state for discussion and comparision.

Additional comments

he abstract could benefit from providing more information on the sample size, participant characteristics, and potential limitations of the study design.

---

## Round 0.2 · Major Revisions

Revisions are still necessary before the manuscript is suitable for publication.

·

Basic reporting

One minor point - please change vision acuity to visual acuity throughout.

Experimental design

Thanks for clarifying the scoring of the visual acuity, where it was scored per line where three or more are correct. This is helpful to know, but I can’t see this in the revised manuscript. I presume in the explanation provided by the authors there is an error where it says if one optotype in the 0.8 row was correct it was scored as 0.6+1 – presumably it should be 0.5, not 0.8.

The authors state that they only present the Frisby once at each level, resulting in a 25% chance of a lucky guess, and said this approach aligns with clinical practices. However, the instructions for Frisby state that there should be repeated presentations when determining the stereoacuity measurement. By choosing to only present once you have increased the variability in the results which needs addressing in the discussion.

Validity of the findings

Original comment - Line 305 – the authors evaluate the significant differences between the natural and dichoptic states, however they have simply fixed on the low p value without considering the magnitude of the differences. Given that the variance has been reported (eg see Bohr I, Read JCA. Stereoacuity with Frisby and Revised FD2 stereo tests. PLoS One. 2013;8.) to be greater than 10” then a median difference of 10” is not clinically significant.
Response:Our selected participants are young medical staff, chosen for their quick responsiveness, strong comprehension skills, and professional expertise. They have VA is within the normal range. However, this approach does have limitations, as it introduces a significant bias due to differences between this group and the broader population. Our study aims to explore the differences between natural and dichoptic states rather than focusing on a diverse demographic that includes children, the elderly, or less cooperative individuals. We aim to leverage the strengths of this particular age group, where stereopsis is typically at its best.
Saladin's research (Saladin IJ. 2005. Stereopsis from a performance perspective, Optom Vis Sci 82:186-205) indicates that stereopsis is optimal in individuals aged 22-29 and slightly diminishes in the 30-39, 40-49, and 50-59 age groups, with a significant decline observed at 60-68 age. Our approach aims to minimize confounding factors by focusing on individuals with the best VA and quickest response times, allowing us to compare the two methods for any discernible differences. Our study intends to qualitatively rather than quantitatively, clarify this issue. This is because the small sample size of our participants may render non-parametric testing for quantitative data less reliable. Additionally, the stereopsis effect resulting from induced vision degradation could differ numerically from actual conditions. Consequently, our study primarily emphasizes identifying significant qualitative differences rather than specific quantitative measurements.

Response to authors - I understand the rationale for the choice of participants and it would lead to the lowest test retest variability, but there still is variability on repeated testing. Psychophysical measures of visual function have some degree of variability in the results, this needs to be considered by the authors and to differentiate between statistically significant differences and clinically significant differences.


Original reviewers comment - The discussion section could be more concise. Some of the content can be placed in the background section. And it is suggested that the authors include other electronic stereoscopic measurement methods under dichoptic state for discussion and comparison.
Response:Thank you for your feedback. Following your suggestion, we have streamlined the discussion section and relocated some of its content to the background section for better coherence and clarity.
Response to authors - This response doesn’t address the reviewers final suggestion in relation to discussing other electronic stereoscopic measurement methods under dichoptic state.

Additional comments

The addition of more participants has changed some of the results to be statistically significant now, showing the importance of the sample size. It also suggests that an even bigger sample size would be beneficial, however there is no mention of a sample size calculation to know what an optimum sample size would be.

In the first review I asked - As the test is modelled on the FNS are there any intellectual property issues related to the use of the principles of the FNS or was permission sought from Professor Frisby? There’s nothing mentioned in the acknowledgements.
Response:Thank you for your constructive feedback. In response, we have included an acknowledgment to Professor Frisby in the Acknowledgments section of our paper.

The inclusion of an acknowledgement is helpful but it doesn't address the question about whether there are any intellectual property issues as the test is copyrighted.

---

## Round 0.3 · accepted · Accept

I have now had the opportunity to read your revised manuscript, and your responses to the reviewer comments. I believe that you have addressed the concerns raised, and I am happy to accept your manuscript.

·

Basic reporting

No comment

Experimental design

No comment

Validity of the findings

No comment

Additional comments

The authors have addressed the comments appropriately.